# Interventions for Preventing Residential Fires in Vulnerable Neighbourhoods and Indigenous Communities: A Systematic Review of the Literature

**DOI:** 10.3390/ijerph19095434

**Published:** 2022-04-29

**Authors:** Samar Al-Hajj, Ediriweera Desapriya, Colleen Pawliuk, Len Garis, Ian Pike

**Affiliations:** 1Department of Epidemiology and Population Health, Faculty of Health Sciences, The American University of Beirut, Beirut P.O. Box 11-0236, Lebanon; 2British Columbia Injury Research and Prevention Unit, British Columbia Children’s Hospital Research Institute, Vancouver, BC V6H 3V4, Canada; ediriweera.desapriya@bcchr.ca (E.D.); cpawliuk@bcchr.ca (C.P.); lwgaris@outlook.com (L.G.); ipike@bcchr.ca (I.P.); 3School of Culture, Media and Society, The University of the Fraser Valley, Abbotsford, BC V2S 7M8, Canada; 4Department of Pediatrics, Faculty of Medicine, The University of British Columbia, Vancouver, BC V6H 3V4, Canada

**Keywords:** residential fires, indigenous communities, fire prevention interventions, fire-related injuries, fire safety skills, safety behaviours

## Abstract

Globally, residential fires constitute a substantial public health problem, causing major fire-related injury morbidity and mortality. This review examined the literature on residential fire prevention interventions relevant to Indigenous communities and assessed their effectiveness on mitigating fire incidents and their associated human and economic burden. Electronic databases including MEDLINE, EMBASE, CENTRAL, and Web of Science Core Collection were reviewed for studies on fire prevention interventions published after 1990 and based on the 4E’s of injury prevention approaches (Education, Enforcement, Engineering, and Engagement). The grey literature and sources including indigenous organizational websites were also searched for eligible studies. Two authors independently screened, selected, and extracted data, in consultation with experts in the field. Outcomes measured included enhanced safety knowledge and practices, decreased residential fires incidents, reduced fire-related injuries and deaths, and lowered costs for healthcare needs. After removing duplicates, screening titles and abstracts, and assessing full texts, 81 articles were included in this review. Of the included studies, 29.1% implemented educational interventions within a variety of settings, including schools, community centres and homes, and included healthcare professionals and firefighters to raise awareness and the acquisition of fire safety skills. Engineering and environmental modifications were adopted in 20.2% of the studies with increased smoke alarm installations being the leading effective intervention followed by sprinkler inspections. Moreover, engagement of household members in hands-on safety training proved to be effective in enhancing household knowledge, fire safety decisions and practices. More importantly, effective outcomes were obtained when multi-faceted fire safety interventions were adopted, e.g., environmental modification and educational interventions, which together markedly reduced fire incidents and associated injuries. This review reveals the dearth of fire prevention evidence gathered directly within Indigenous communities. Nonetheless, relevant fire prevention recommendations can be made, calling for the adoption of combined and context-sensitive fire prevention interventions tailored to targeted Indigenous and vulnerable communities through multiple approaches and measures. Follow-ups and longitudinal studies are critical for accurate evaluation of the long-term outcomes and impacts on preventing residential fires.

## 1. Introduction

Residential fires account for a sizable proportion of fire incidents globally, highlighting its significance as a major public health problem [1]. Fire-associated morbidity and mortality represent the fourth most common cause of unintentional injuries, affecting millions of lives worldwide [2]. Estimates from the Global Burden of Disease study (GBD 2019) reported nearly 110,000 fire-related deaths globally in 2019 [3]. Further to its mortality burden, fire-related injuries are associated with prolonged hospitalization and lifelong disfigurements that equally impact the injured persons’ physical and emotional well-being [2]. Moreover, the high costs of residential fires exceed the damage to residential properties and are estimated to be nearly 10 times higher than the actual reported costs [4].

Residential fires disproportionally affect vulnerable communities owing to their complex relationship with their surrounding environment and the underlying socioeconomic characteristics of households [5]. Several studies examined the increased risk of residential fires particularly among clustered and overpopulated communities, underprivileged occupants of older houses, residents of buildings with sub-standard fire safety measures, and displaced individuals [6,7,8,9,10,11]. Available research evidently demonstrated the link between community characteristics and higher frequency of residential fires. Indigenous communities are particularly prone to heightened risks of various types of unintentional injuries including fire-related morbidity and mortality [12,13] due to multiple factors such as socioeconomic status, overcrowded living conditions, and limited access to healthcare services in rural locations [5,12,13].

Numerous preventive strategies and interventions have been developed and implemented to protect individuals against residential fires. These interventions included fire education programs (knowledge on common causes of preventable fires), home visitations and inspections (fire-safety hazard, smoke alarm and sprinkler installation) and fire prevention legislations [14,15]. Specific interventions have targeted high-risk groups such as vulnerable individuals, young children, the elderly, and the youth (e.g., juvenile fire-setter programs). The longitudinal effectiveness of existing fire prevention interventions varies considerably in terms of enhanced residential fire safety, reduced frequency of fire incidents, and more importantly, decreased fire-related injuries and casualties. Nonetheless, existing fire prevention initiatives lack conclusive and formal evaluation as to their effectiveness and success, particularly in the longer term.

This review aims to systematically examine the fire prevention literature and identify existing evidence-based interventions and practices designed and developed to reduce the risk of residential fires and associated injuries and deaths. The study seeks to evaluate and synthesize the findings, making them available for the uptake of relevant interventions in vulnerable neighbourhoods and Indigenous communities. Evidence from this review will improve understanding and provide insights into the effectiveness of existing fire prevention initiatives, stimulate improvements of public health policies in residential fire prevention efforts, and guide the implementation of data-driven, research-based approaches to reduce fires, particularly in Indigenous communities and across similar settings.

## 2. Methods

The fire prevention literature was examined, and evidence-based effective fire prevention initiatives were compiled. This literature was reviewed and categorized through the lens of the ‘4 E’s of injury prevention’ (4 E’s): Education (e.g., educating individuals about changing behaviours), Enforcement (e.g., safety legislation and policies, including passing, strengthening, and enforcing voluntary standards, regulations, and laws to suppress residential fire), Engineering (e.g., designing, developing, and manufacturing products to build safer environment), and Engagement (e.g., engaging stakeholders in the process of systemic change and safety promotion). All reports published after 1990 and all types of intervention designed to reduce the risk of residential fires and fire-related injuries were included in this review. Protocols of this study were published in the International prospective register of systematic reviews PROSPERO.

### 2.1. Search Strategy

The search strategy was developed with the support of a health sciences librarian (CP). The following electronic databases were searched: MEDLINE, EMBASE, CENTRAL, Web of Science Core Collection, Trials Register of Promoting Health Interventions (TRoPHI), PAIS Index, Educational Resource Information Center (ERIC), FireDOC (National Institute of Standards and Technology), and IEEE Xplore Unpublished studies. Grey literature was searched using the following sources: Theses and dissertations (ProQuest Dissertations and Theses Global, The Networked Digital library of Theses and Dissertations—NDLTD), conference proceedings (Papers First and Proceedings via WorldCat FirstSearch), government reports (OpenGrey, Grey Literature Report), and Indigenous organizational websites (Training Programs in Epidemiology and Public Health Interventions Network—TEPHINET) (refer to Appendix A for the search strategy). Relevant systematic reviews were included [16,17,18,19,20] to add any studies identified through recursive searches of their reference lists. Furthermore, Google Scholar was used to review the citing references for all included studies and reports and any missed literature was added. Moreover, relevant fire safety materials included on Indigenous organization websites were searched for published and unpublished literature.

### 2.2. Eligibility Criteria

Articles were included in the review if the study evaluated a fire intervention and demonstrated its impact on the following outcomes: (1) Knowledge/Attitude/Behaviours (KAB) improvement, (2) Reduction of residential fires risk/incidence/frequency, (3) Lower injuries/hospitalizations/deaths rates, (4) Safety enhancement in the extent of infrastructure damage and fire suppression, and (5) Lower costs for healthcare needs/health response.

### 2.3. Data Screening, Selection and Extraction

Articles revealed by the search were exported to the Covidence software platform [21]. Duplicates were removed and screening of titles and abstracts, followed by full-text reviews, was performed by two independent authors (SA, ED). We adopted the PICOS (Population, Intervention, Comparators, Outcomes, Study Design) framework as a systematic review approach to screen and select eligible studies and include them accordingly. Descriptions such as ‘Fire prevention’, ‘Residential fire’, ‘Intervention’, ‘Indigenous’ were used to retrieve eligible reports. All identified citations were evaluated based on the inclusion criteria.

Screening of titles and abstracts was performed independently and in duplicate by two authors (SA, ED) to select potentially eligible studies. Following the identification of eligible articles, two authors (SA, ED) independently evaluated the full text of relevant articles and abstracted data. Disagreements were resolved by discussion or by consultation with a third reviewer (IP) in order to reach consensus. Abstracted data include study author(s), year of publication, country, fire intervention adopted, behaviour addressed, population, setting, and the intervention outcome. Reported studies published after 1990 were added to the review though hand searching and examination of the reference list of existing reviews [18,19,20].

### 2.4. Data Analysis and Synthesis

All results were subject to double data entry by two authors (SA, ED). The effectiveness of existing fire prevention initiatives was investigated based on the 4E’s of injury prevention intervention approaches. Subgroup analyses were performed for the 4E’s fire prevention measures (Education, Engineering, Enforcement, and Engagement) and their effectiveness level in reducing fire incidents risks and fire-related injuries and deaths. Due to the high heterogeneity of included interventions, populations, and their outcomes in the studies included, we were unable to group the data, and, therefore, unable to include a meta-analysis.

Randomized and non-randomized (including cohort and case-control) studies were assessed for methodologic quality using the Downs and Black Checklist [22]. This checklist includes 27 items, widely covering areas reporting quality, external and internal validity, (bias and confounding), and the study power [23,24,25]. Maximum scores of 27 and 26 were available for randomised and non-randomised studies, respectively. The quality of each study was independently assessed by two authors, with discrepancies resolved through discussion and consensus. Using the Downs and Black Checklist, the included studies scored between 13 and 20.5 points for quality, out of a possible 27 points. The included studies were of low to moderate quality.

Methodological quality of systematic reviews was assessed using the critical appraisal tool, A Measurement Tool to Assess systematic Reviews (AMSTAR2-Bruyère Research Institute, Ottawa, ON, Canada). AMSTAR 2 is an empirically derived, reliable, validated 16-item critical appraisal tool used to assess the methodological quality of systematic reviews of RCTs and epidemiological studies [26]. Quality of the reviews was calculated using the checklist form. We considered an AMSTAR score: high (none or one non-critical weakness), moderate (more than one non-critical weakness), low (one critical flaw with or without non-critical weaknesses), and critically low (more than one critical flaw with or without non-critical weaknesses), respectively [26].

Covidence was used to manage the retrieved studies, review abstracts, screen for inclusion/exclusion, extract data, and perform bias assessment risk. Tableau software was adopted to visualize the studies outcomes data. The Adobe Illustrator tool was used to design the chord chart (Adobe, San Jose, CA, USA).

## 3. Results

### 3.1. Study Characteristics

The search yielded a total of 5806 records. After identifying and removing duplicates, 3044 unique records were included. Title and abstract screening resulted in the exclusion of 2617 records, and the remaining 427 articles were assessed for eligibility based upon the outcomes of interest in this review, with a further 287 articles excluded, leaving 140 articles for full text assessment. Six articles could not be retrieved, and 77 articles were excluded for reasons as articulated in Figure 1. Sixty-three articles were included in the review [1,2,3,4,5,6,7,8,9,10,11,12,13,14,15,27,28,29,30,31,32,33,34,35,36,37,38,39,40,41,42,43,44,45,46,47,48,49] together with an additional 18 articles retrieved through reference list reviews, resulting in a final total of 81 articles included in this review.

Forty-two of the 81 articles originated in the USA (51.8%), 14 (17.2%) in the UK, eight (9.8%) in Canada, seven (8.6%) in Australia, four (4.9%) in Sweden, two (2.4%) in New Zealand, and one article in each of Japan, Germany, France, and Iran. The majority of articles reflected Randomized Control Trials (RCTs) (32%), followed by systematic review (16%), qualitative research (12%), non-randomized experimental studies (10%), cohort studies (8%), cross sectional (6%), and other study designs (16%).

### 3.2. Interventions Sub-Analysis

The retrieved studies were classified based on one or more of the 4E’s of injury prevention outcome that was targeted: Education (*n =* 23, 29.1%), Engineering/Environmental changes (*n =* 16, 20.2%), Enforcement (*n =* 6, 7.5%), and Engagement (*n =* 5, 6.3%). Studies focused on more than one of the 4E’s included: Educational and Environmental modifications (*n =* 17, 21.5%), Engagement and Environmental modifications (*n =* 7, 8.8%), and Education, Environment, and Enforcement (*n =* 5, 6.3%) (Figure 2).

Interventions were implemented within a variety of settings, such as schools, homes, community centres, nursing homes, and clinics. Most studies initiated a preliminary needs assessment to address a local residential fire problem, identified high-risk populations, and mapped community households prior to intervention implementation and data collection.

Educational Interventions: The majority of included studies (*n =* 23, 29.1%) adopted educational interventions mainly related to smoke alarms installation and maintenance, fire escape plan development, and fire guard utilization [1,14,15,20,25,31,35,36,39,47] (Figure 2). Educational materials (brochures, pamphlets, or posts on social media platforms) were disseminated through door-to-door fire safety campaigns, community safety programs and child healthcare counselling at schools, nursing homes, clinics, and medical centres [9,50,51] aimed to enhance individual knowledge and fire safety skills, behaviours, and practices acquisition, and ultimately to prevent residential fire and associated injuries [15,20,25,29,31,32,38,39,47,48,51,52,53]. These interventions mainly targeted at-risk groups within vulnerable neighbourhoods including parents of young children, elderly, and low socio-economic households [25,29,31,32,39,47,48,51,54,55,56]. Some studies recruited healthcare workers, firefighters, community leaders, and volunteers to support educational intervention implementation [43] and foster fire safety skills and attitudes among youth [29,52] through Cognitive Behavioural Therapy (CBT). Healthcare worker involvement raised access to households by 21%, and subsequently the prevalence of smoke alarm installation [17]. Although educational interventions improved knowledge and changed behaviours, limited evidence was reported on their influence in reducing fire incidence [29,52] and injuries [8,57,58] and more importantly, on sustaining the long-term safety knowledge and practices [59] among adults [32] and elderly [31] populations, which necessitated follow-ups and safety concepts reinforcement.

When strengthened by the provision of free or reduced-cost safety equipment, fire educational interventions had a positive impact on enhanced safety behaviours and practices [12,14,28,35,48,50,54,58,60], though a limited effect on preventing fire incidence [9]. Counselling of parents on home fire safety and the importance of a functioning smoke alarm, coupled with the provision of low-cost safety equipment, led to added installation and maintenance of operational smoke alarms among the intervention groups [50,58,61]. Although small in number, there was evidence that educational interventions that included access to safety equipment led to a significant reduction in fire incidence, associated injuries and deaths, and economic saving realized by individuals and the healthcare system [12,53].

Engineering and Environmental Modifications: Sixteen studies (20.2%) modified the existing environment to reduce the risk, and severity of residential fire incidents, enhance household safety, and prevent fire-related injuries and fatalities [6,10,18,24,34,36,41,42,57,62,63,64,65,66,67,68], including smoke alarm installation [24,34,57,65,68] and sprinkler inspection (Figure 2). The Centers for Disease Control and Prevention (CDC) recommends having a working smoke alarm with a long-lasting lithium-ion battery on every level of a home, as functioning smoke alarms reduce house fire death risk by 50% [2,12,24,33,57,65]. Inspecting fire alarms and checking batteries regularly is the most prominent challenge to ensure sustainable safety and protection among households. Sprinklers surpassed smoke alarms in decreasing fire-related morbidity and mortality leading to a reduction of 100% in fire fatalities and 72% in property damage [6,62,63,64], particularly given that sprinklers contain fire to its ignition origin in up to 97% of the cases [6,33,57,63,64]. Fire resistant clothing, beds, and sofas were effective in reducing fire-related injuries [35]. Smoke alarm giveaway programs with direct home visits by community leaders, fire safety inspectors, local firefighters, and community volunteers for installation, annual battery checks, and educational materials provision were the most effective environmental modifications within vulnerable neighbourhoods but had a limited long-term effect [10,18,54,55].

Combining environmental modification (e.g., smoke alarm installation, sprinklers inspections) and educational interventions demonstrated substantial impact and greater benefits in reducing fires and preventing injuries [2,5,7,8,9,17,19,22,43,44,55,69,70,71,72]. The primary outcome for these multi-faceted fire intervention programs was the reduction in fire incidents and associated injuries and deaths, and the mitigation of costs to the healthcare system. Smoke alarm installation coupled with fire safety education and supported by community members led to increased effectiveness in the acquisition and adoption of the safety changes [17]. Environmental modifications strengthened with home visits by public health professionals, community partners, and firefighters indicated persistent positive effects on enhancing environmental safety and changing behaviours in the community, leading to reduced fire incidence particularly among vulnerable populations [7,17,54,55,59,70]. Moreover, environmental modifications and education intervention programs proved to be cost-effective [68] and may have strong relevance for Indigenous communities due to similarities in socio-economic status and in environmental context of the populations under investigation. A study conducted among an Indigenous community in New Zealand demonstrated the substantial impact of education and smoke alarm installation in reducing injury hospitalization rates [5].

Enforcement: Six studies (7.5%) assessed existing fire safety laws and regulations implemented to prevent residential fires and their associated injuries [1,3,4,13,21,30] (Figure 2). Mandatory smoke alarm installation across German federal states led to a substantial reduction in fire incidents and fire-related mortality and morbidity [13]. An Australian study showed the link between enforced smoke alarm installation and the reduction of fire injury, hospitalization, and deaths rates [21]. Moreover, Laing and Bryant confirmed the reduction of child injury rates in a study conducted in New Zealand following ‘safe clothing’ legislation that prevented and protected children from nightwear fire incidents. While some enforced fire regulations reduced fire related injuries, others reported little evidence of achieving any of their intended fire prevention outcomes. Bonander et al. showed minimal to null effect of the Fire Safe Cigarette Law (FSCL) enforcement, and the banning of all Non-Reduced Ignition Propensity (Non-RIP) cigarettes, on reducing fire occurrences or fire mortality and morbidity [3,4]. On the contrary, Alpert et al. (2014) demonstrated the impact of the FSCL in reducing residential fires with a small sample size study [1], but noted that a larger sample size is required to confirm a more definitive conclusion.

Engagement: Five studies (6.3%) proposed fire interventions that interact with participants and engage with community leaders, firefighters, healthcare workers, and social workers on discussions related to fire safety [46], narrative simulation of fire emergency situation [73], or hands-on training of fire safety skills [71,74,75] (Figure 2). These efforts led to higher acceptance and commitment of participants [60] and more importantly improved safety knowledge and enhanced ability to identify hazardous behaviours and address fire safety deficiency among participants in their environment [23,45,73,74,75,76]. Individual engagement in hands-on-training with fire service personnel and firefighters through home-based face-to-face visits helped households adopt effective fire safety measures and emergency responses to mitigate fire risks, gained through enhanced safety skills, particularly among vulnerable populations, reducing occurrence by some 5% [74,76]. One study ensured sustainable safety behaviour practices with fire safety pledges from participants [69,71,74]. One study indicated that the intervention cohort that received educational materials and hands-on safety training were 16% more likely to install and maintain a functioning smoke alarm compared to those who only received educational interventions [71].

Seven studies confirmed that participant engagement in safety behaviours accompanied with environmental modification interventions resulted in a higher impact on various outcomes including enhanced safety and reduced fire-related injury [2,11,16,33,46,59,71]. Tailored and culturally-appropriate intervention programs and strategies (i.e., appropriate interventions tailored to child age, mothers language, parents cultural background) were successful in engaging targeted populations with a clear impact on their knowledge improvement, fire safety behaviours, and the reduction of fire-related injuries [59].

Education, Environmental Modifications, and Enforcement: Five previous reviews examined universal and distinctive resident-, property-, and fire-related risk factors associated with residential fires and thoroughly described effective interventions implemented in different countries [14,15,77,78,79], including the UK, USA, Canada, Europe, and Japan (Figure 2). These reviews showed that households with young children, older adults, individuals with physical and mental disabilities, smokers and individuals with alcohol and drug addictions, single-family households, and low-income families were all at particularly heightened risk of fire-related injuries and deaths. Moreover, these reviews synthesized preventive strategies, methods, and best practices to prevent residential fires including environmental modification, promotion of safety regulations, and changes in risk behaviour among individuals. Proven best practices can be adopted as successful interventions to stimulate improvements in fire prevention practices in high-risk communities, with further adaptation to communities’ unique environmental, cultural, and social factors.

Others: Some recent interventions were theoretical in nature, computer simulations, and laboratory experiments that utilized technology—automated systems, artificial intelligence, fuzzy logic, and machine learning—and hypothetically reduced residential fires and enhanced fire safety and fire-injury prevention methods, including the use of a firefighting robot, electronic smoke detector nose (PEN3), and Geospatial Information System (GIS) applications [79,80,81,82,83,84,85,86,87,88,89,90,91,92]. Adopting the Internet-of-Things (IoT), wireless sensors, and advanced modelling in automated fire detection networks [93,94,95,96,97,98] represents a critical step towards enhancing residential fire safety and advancing the research field of fire prevention going forward [56,87,91,99,100,101,102,103,104,105,106,107,108,109,110,111,112,113,114,115,116,117,118].

## 4. Discussion

To the best of our knowledge, this review is the most comprehensive study to date. It examined the literature related to residential fire safety and prevention interventions in vulnerable neighbourhoods and Indigenous communities. It further synthesized the evidence of the effectiveness of various interventions based on outcomes impacting the reduction of fire-related injuries and deaths, the decrease in fire incidence, and the improvement in fire safety knowledge and practices, categorizing them within one or more of the 4E’s of injury prevention—Education, Engineering and Environment, Enforcement, and Engagement. Although retrieved reports showed a dearth of studies specific to Indigenous communities, efforts were made to review and synthesize the existing fire prevention literature, noting the potential applicability to Indigenous populations.

Multiple factors, including environmental, behavioural, and social aspects contribute to individual risk of fire-related injuries and deaths [97]. Available studies identified these important determinants including maternal education, socioeconomic status, single-parent household, absence of adequate adult supervision along with the lack of fire escape plans, and smoke alarm functionality [77,96,97]. This review underscores the effectiveness of fire interventions in reducing fire related morbidity and mortality and their impacts on individuals and the healthcare systems. One systematic review and meta-analysis revealed that installed and functioning smoke alarms reduce death rate per fire incident by approximately half [65].

Consistent with previous research, this review reveals the lack of evidence specific to the effect of educational interventions alone on the reduction of fire-related injuries and deaths [8,57,58,59] compared to other interventions such as environmental modification (smoke alarms installation and maintenance) [2,12,24,33,57,65] and enforcement [13]. Moreover, this review confirms that combining multiple interventions including education, the provision of safety equipment, home inspection, and proactive engagement of household members yielded the most effective outcomes and represented best fire safety prevention practices in the current research literature. Integrated multiple aspects of the 4E’s interventions led to a more prominent outcome in terms of improving knowledge, changing behaviours, and more importantly, preventing fire-related injuries and deaths. Tailoring fire interventions to population demographics, age, setting, and types of residential homes resulted in more successful outcomes [35,41,42].

Although a considerable amount of fire prevention research has targeted vulnerable communities and at-risk populations, there is a dearth of literature specific to Indigenous communities globally [5,70]. Existing research on Indigenous populations to date are mainly epidemiologic studies that described and examined the burden and aetiology of residential fire-related injuries. Given the higher prevalence of residential fire among vulnerable populations, single-headed households, individuals living in lower socioeconomic strata, homes with non-functioning smoke alarms and older housing construction [2,119], Indigenous communities with similar profiles and characteristics might, therefore, be considered at higher risk of residential fires and related injuries. This review demonstrated that several fire interventions are applicable to the Indigenous community and can be adopted to alleviate the repercussions of residential fires on vulnerable populations.

Using the AMSTAR 2 Checklist [26], the methodologic quality of the included systematic reviews ranked between critically low and moderate. None of the included studies ranked high, including Cochrane systematic reviews. The overall results demonstrate very low confidence in the results of most included systematic reviews of residential fire safety enhancement interventions. Public health and policy makers charged with the delivery of the most efficacious interventions to high-risk households/communities are encouraged to attend to the critical appraisal of previous systematic reviews using the AMSTAR 2 Checklist before making their decisions [26].

This review comprehensively examined and synthesized existing fire intervention measures and their implications on reducing fire incidents and associated human and economic impacts. Nonetheless, future research warrants the close examination of the framework utilized in these interventions to inform the design and development of proven-effective fire interventions and tailor them to the needs and existing resources of the targeted communities.

A key goal of this review was to make recommendations aimed at Indigenous communities. Given the low to moderate quality of previous systematic reviews included, together with the dearth of Indigenous-specific research, we nonetheless offer a series of recommendations that can be considered when developing and implementing specific fire safety interventions for Indigenous communities:***Operationalized fire prevention research***: The increased risk of fire-related injuries and deaths among Indigenous populations highlights the significance of fire prevention programs. This review demonstrated the clear dearth of evidence on fire prevention initiatives specific to Indigenous communities. Although some Indigenous communities are at higher risk than others, evidence from this research can be adapted and implemented within these communities to engage and lead initiatives that will mitigate the burden of fire-related injuries and deaths.***Combination of multiple E approaches for an impactful intervention***: Fire interventions that combine multiple measures (e.g., providing fire safety education, installing smoke alarms, distributing free, low cost or discounted safety equipment, education, etc.) along with engaging household members were more effective at improving safety practices and reducing fire injuries and deaths. The design of future fire interventions should rely on multiple approaches including mass media campaigns, education, clinician counselling, home visits and inspections using neighbourhood canvassing methods, fire alarm installation and timely battery change reminders, and legislation.***Tailored intervention is essential to lead a significant impact***: Synthesized evidence demonstrated the importance of culturally sensitive and context-specific interventions and their effectiveness when tailored to the unique characteristics of the targeted population, particularly higher-risk communities [58]. The provision of safety items coupled with a home visit tailored to child age and maternal culture was shown to be an effective intervention in a hard-to-reach population. Creating interventions in the language of the target population is desirable to engaging individuals and to increase self-efficacy for safety behaviours. Affordability of fire prevention measures (clothing, furniture, fire alarms) is critical to enhance adoption of fire safety behaviours.***Long term assessment of intervention outcomes***: Evaluation of injury prevention programs is critical for assessing the program strategies and measuring its effects on reducing injury related morbidity and mortality and for refining it with the aim to increase its likelihood of achieving successful outcomes. A multi-dimensional approach and long-term post-intervention evaluation should be adopted to assess the expected fire injury outcome. Few studies reported post-intervention fire-related injury and death reductions. Long-term evaluation in these educational programs was accurately enabled with access to hospital surveillance data besides its impact on fire morbidity and mortality [53]. It is crucial to conduct longitudinal studies to evaluate the long-term outcome of implemented interventions and their effectiveness and impacts on preventing residential fires and injuries.

Despite the extensive search strategy and review process adopted, this review is subject to some limitations. Firstly, a source of publication bias might exist given this review was exclusive to scientific manuscripts and published grey literature/technical reports without including fire intervention posters, abstracts, or unpublished studies. This might have affected the number of studies included in each intervention category. Secondly, the dearth of studies related to Indigenous communities might affect the generalizability of this review to Indigenous communities.

## 5. Conclusions

This paper presents a comprehensive review of the fire intervention literature and synthesized evidence on the effectiveness of various interventions in preventing residential fires and reducing their associated injuries and deaths. This review reveals the dearth of fire prevention evidence gathered directly within Indigenous communities. The included studies proposed a combination of effective interventions (e.g., educational, engagement, enforcement, and environmental modifications) that can be concurrently implemented to achieve successful and impactful outcomes. Future research on the impact of residential fire-safety and prevention intervention specific to Indigenous communities is warranted.

## Figures and Tables

**Figure 1 ijerph-19-05434-f001:**
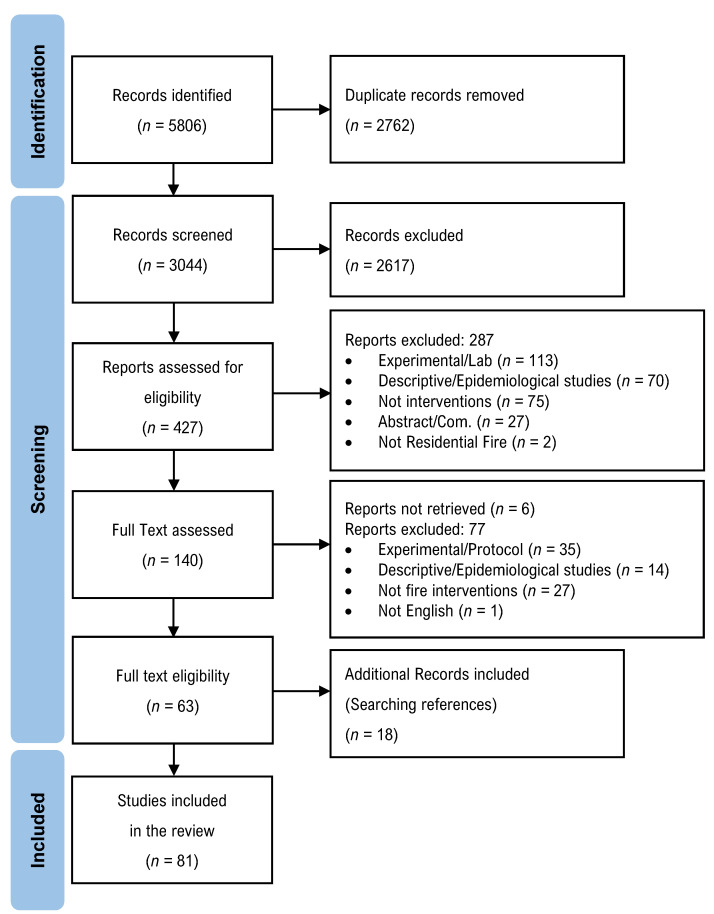
Preferred Reporting Items for Systematic Reviews and Meta-Analyses (PRISMA) flow diagram.

**Figure 2 ijerph-19-05434-f002:**
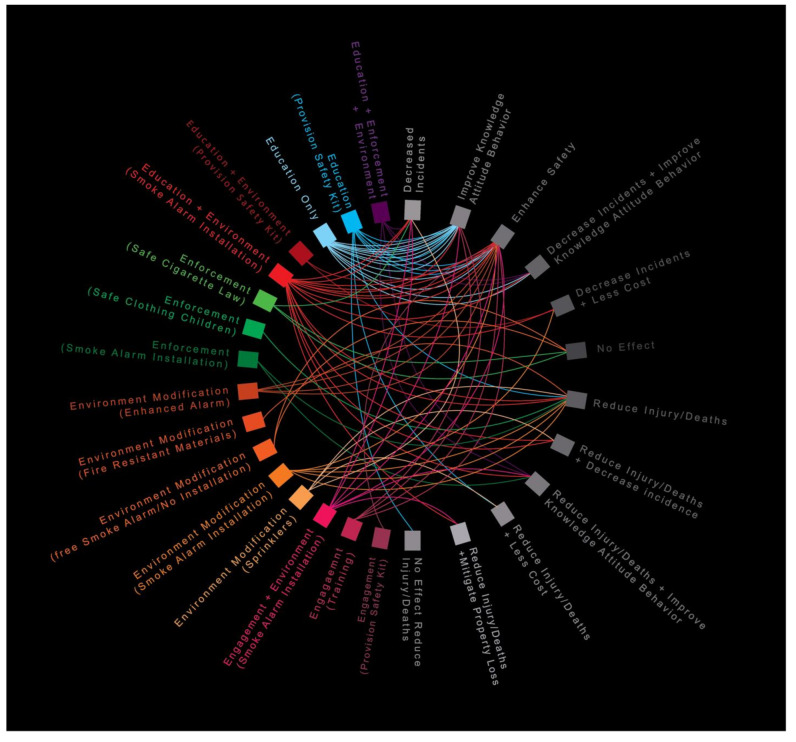
Chord chart links each of the 4E’s interventions with its measured outcome(s) as reported in the included studies.

## Data Availability

Data were compiled though Covidence (https://www.covidence.org/) and will be available upon request.

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
