# Peer review of "Interventions for Preventing Residential Fires in Vulnerable Neighbourhoods and Indigenous Communities: A Systematic Review of the Literature"

_ijerph, 2022, doi:10.3390/ijerph19095434_

Round 1

Reviewer 1 Report

This paper summarizes the literature on fire prevention and intervention in vulnerable neighborhoods and indigenous communities, which found that educational, engineering and environmental interventions are effective for fire prevention. The research is of great significance for the guidance of fire prevention. It can be accepted if the following question be answered. 

Figure 2 would be better if it could explain what software was used to draw it.

Author Response

We would like to thank the reviewers for their feedback and insightful suggestions to modify the manuscript. Please find enclosed our point-by-point responses in blue and italic:

Reviewer 2 Report

I would like to congratulate the authors on taking an intersectional approach to reviewing the evidence of a very important topic. Nevertheless, I think that there are some weaknesses in the paper which you need to address. You state that there is limited evidence which relates specifically to indigenous communities, but the value of such a review is to offer explanation and to make more concrete suggestions than the generalities you make in several places such as: ‘proven best practices can be adopted as successful interventions to stimulate improvements in fire prevention practices in high-risk communities, with further adaptation to communities’ unique environmental, cultural, and social factors’. Yet you offer no explanation of these ‘unique factors’ – which would help a reader of your paper who might wish to make use of your findings. Indeed, when I read the recommendations they seem like common sense and could have been produced without your review.

Below I give some suggestions where I think improvements are needed.

P9 lines 335-7. ‘One systematic review and meta-analysis revealed that installed and functioning smoke alarms reduce death rate per fire incident by approximately half’ – my reaction here is, on the one hand any life saved is good, but on the hand ‘only half’! Why?

How do you define (and so identify) ‘vulnerable communities’? I am not clear about the search terms you used related to the population. In the paper, you use the term ‘indigenous communities’ – this term would not be applicable in all contexts, for example, the UK would use ‘ethnic minorities/groups/communities’. There is certainly academic research on this topic in the UK – indeed I understand that the UK might even have official data related to fire disaggregated across ethnicity.

It’s not clear as to whether or people are owner-occupiers or tenants – the responsibility and capacity to act on fire prevention is different. Are the buildings single or multiple household occupancy?

Is the educational material only in the majority/official language of the country? Is this not problematic for minorities – including indigenous communities? On page 10 lines 401-2 you actually have a recommendation related to this – although not citing any evidence. (Creating interventions  in the language of the target population is desirable to engaging individuals and to increase self-efficacy for safety behaviours).

P7 line 240-1 You make the statement “Fire resistant clothing, beds, and sofas were effective in reducing fire-related injuries”. These are possibly more expensive than non-fire resistant items – so are more likely to be out of purchasing capacity of low-income households. I’d add fire alarms to this list as well.

Line 252 - Environmental modifications – please explain.

Lines 254-255 “leading to reduced fire incidence particularly  among vulnerable populations” – why? what was done to produce this effect? It is possible that the authors you are citing don’t explain – but you should then mention this!

Lines 256-258 “environmental modifications and education intervention programs proved to be cost-effective and may have strong relevance for Indigenous communities”. It’s not clear what point you are making here. So more explanation would be helpful!

P8 lines 293-4 mentions the success of ‘Tailored and culturally-appropriate intervention programs and strategies’ – can you give an example of what this means?

Author Response

(The authors gave the same response as above.)

Reviewer 3 Report

The manuscript (MS) presents a review of interventions for preventing residential fires in vulnerable neighborhoods and indigenous communities. The most important electronic databases were searched for evidence of fire prevention measures regarding the Education, Enforcement, Engineering, and Engagement, i.e., the 4E’s of injury prevention approaches. Outcomes included improved safety knowledge and practices, decreased residential fire frequencies, reduced fire-related injuries and deaths and reduced costs for healthcare. 81 eligible studies were analyzed. Were adopted, multi-faceted fire safety interventions, e.g., environmental modification and educational interventions, together resulted in the most effective outcomes regarding fire incidents and fire related injuries.

Title

It is representative for the study. (Maybe consider substituting "Evidence" with "Literature".)

Abstract

Consult the IJERPH Template regarding removal of sub-headings, i.e., Introduction, Methods, Results and Conclusions.

Line 19: Use "author/authors" when describing you, i.e., not "reviewers". Check throughout.

Line 22: tiles ?

  1. Introduction

It is short and quite descriptive.

Consult the IJERPH Template regarding referencing, i.e., brackets [ ].

  1. Methods

Line 92: You need to define what definition of risk you use. Usually, the term risk(s) consists of frequencies and consequences (frequencies and outcomes). Check the wordings in Line 92, i.e., "… risk and frequency …" as risk already bears the frequency within the definition. Check throughout.

Line 95: IJERPH usually like all links to data referred as a reference with the notation "accessed day/month/year", ref. the IJERPH Template.

Line 156: … tool, AMSTAR2, i.e., an empirically … But remember to state manufacturer, town and country for this product, as well as possible other products, throughout.

  1. Results

Line 175 and 177: Rewrite to make it clear whether 77 or 81 articles were included, and whether these were indeed articles or included grey literature.

Figure 1: Top left. What is "Indicatio"? Missing a letter? Lowest right box difficult to read. Use less line spacing to get three full lines? In general, use less line spacing in the figure? When using an abbreviation first time, spell to fully, e.g., PRISMA.

Line 187: Not n=23. Better with n = 23, etc. Check throughout.

Line 203: … (Fig 2) should be … (Figure 2). Check throughout.

Line 226: … there evidence …  ? (was? is?)

Line 230 and 235: see comment for Line 92.

Line 273: Write a proper reference, ref. the IJERPH Template.

  1. Discussion

Line 325: Different from Line 86-90. Which is correct? (In particular the word "Environment".)

Line 328-414 would benefit by some more referencing.

Missing final statement that this is the most comprehensive study to date?

Indicate whether the findings are of general validity, i.e., for more average communities, or outstanding for the communities analyzed.

  1. Conclusions

State the most important finding (and not only that you propose a combination…).

References

  1. Should be: Branche, C.; Ozanne-Smith, J.; Oyebite, K.; Hyder, A.A. World report on child injury prevention. 2008. (Which organization, where (city, country)), etc. Consult the IJERPH Template regarding the list of references throughout.

Add DOIs where available.

Thank you for an interesting read. 

Author Response

(The authors gave the same response as above.)
